# Circulating miR-122-5p and miR-375 as Potential Biomarkers for Bone Mass Recovery after Parathyroidectomy in Patients with Primary Hyperparathyroidism: A Proof-of-Concept Study

**DOI:** 10.3390/diagnostics11091704

**Published:** 2021-09-17

**Authors:** Seunghyun Lee, Namki Hong, Yongnyun Kim, Sunyoung Park, Kyoung-Jin Kim, Jongju Jeong, Hyo-Il Jung, Yumie Rhee

**Affiliations:** 1Department of Internal Medicine, Severance Hospital, Endocrine Research Institute, Yonsei University College of Medicine, Seoul 03722, Korea; shlee2@yuhs.ac (S.L.); nkhong84@yuhs.ac (N.H.); 2Yonsei University Health System, Seoul 03722, Korea; YNKIM@yuhs.ac; 3Department of Mechanical Engineering, Yonsei University, Seodaemun-gu, Seoul 03722, Korea; angelsy88@gmail.com (S.P.); uridle7@yonsei.ac.kr (H.-I.J.); 4Department of Internal Medicine, Korea University College of Medicine, Seoul 02841, Korea; kyokyo0720@gmail.com; 5Department of Surgery, Thyroid Cancer Clinic, Severance Hospital, Yonsei University College of Medicine, Seoul 03722, Korea; JUNGJONGJ@yuhs.ac

**Keywords:** microRNA, primary hyperparathyroidism, bone mineral density

## Abstract

Primary hyperparathyroidism (PHPT) is the leading cause of secondary osteoporosis. Although bone mineral density (BMD) tends to recover after parathyroidectomy in PHPT patients, the degree of recovery varies. Circulating microRNAs (miRNAs) profiles are known to be correlated with osteoporosis and fracture. We aimed to investigate whether osteoporotic fracture-related miRNAs are associated with postoperative BMD recovery in PHPT. Here, 16 previously identified osteoporotic fracture-related miRNAs were selected. We analyzed the association between the preoperative level of each miRNA and total hip (TH) BMD change. All 12 patients (among the 18 patients enrolled) were cured of PHPT after parathyroidectomy as parathyroid hormone (PTH) and calcium levels were restored to the normal range. Preoperative miR-19b-3p, miR-122-5p, and miR-375 showed a negative association with the percent changes in TH BMD from baseline. The association remained robust for miR-122-5p and miR-375 even after adjusting for sex, age, PTH, and procollagen type 1 N-terminal propeptide levels in a multivariable model. In conclusion, preoperative circulating miR-122-5p and miR-375 levels were negatively associated with TH BMD changes after parathyroidectomy in PHPT patients. miRNAs have the potential to serve as predictive biomarkers of treatment response in PHPT patients, which merits further investigation.

## 1. Introduction

Primary hyperparathyroidism (PHPT), a leading cause of secondary osteoporosis, is characterized by hypercalcemia and bone loss owing to the overproduction of parathyroid hormone (PTH) [1,2]. In patients with PHPT, the only curative treatment is surgery [3]. Previous studies reported that the risk of fracture reduced and bone density improved in patients who underwent parathyroidectomy as opposed to patients who did not [3,4]. However, there is a large variance in bone mineral density (BMD) changes after parathyroidectomy among PHPT patients [5,6]. Sharma and colleagues demonstrated that BMD reduced after surgery in 31% of patients with PHPT [5]. PTH alone does not explain all changes in BMD after surgery [5,7]. Although alkaline phosphatase (ALP), type 1 cross-linked C-terminal telopeptide collagen (CTx), and procollagen type 1 N-terminal propeptide (P1NP) have been suggested as predictors of BMD changes after surgery in PHPT patients, these observations remain controversial [6,8,9,10].

In clinical situations, anti-osteoporotic medications are not started immediately after parathyroidectomy in most patients with PHPT. This results from concerns about anti-resorptive or anabolic medications that may interfere with postoperative BMD recovery. Anti-resorptive drugs inhibit and reduce osteoblast activity owing to the lack of osteoclast-derived coupling factors [11]. Moreover, there have been no studies other than with strontium ranelate on the use of anabolic drugs within 1 year after surgery in patients with PHPT [12]. However, considering the high risk of fracture immediately after surgery [13], biomarkers that can predict postoperative BMD improvement may help to preemptively consider anti-osteoporotic drugs.

Recently, microRNAs (miRNAs) and non-coding single-stranded RNA molecules, which are known to regulate gene expression through post-transcriptional modification, have emerged as novel noninvasive disease biomarkers [14,15]. Circulating miRNAs in human plasma are protected from endogenous RNase activity and are stable [16]. In previous reports, miRNAs have been suggested as major regulators of bone metabolism [17,18,19,20,21,22]. miRNAs associated with osteoporotic fractures have also been thought to be associated with bone deterioration [23,24,25]. These osteoporotic fracture-related miRNAs can be predictors of postoperative BMD changes in patients with PHPT.

Thus, in this study, we investigated whether osteoporotic fracture-related miRNAs can predict postoperative BMD changes in patients with PHPT.

## 2. Materials and Methods

### 2.1. Study Design

Among the 164 patients who underwent parathyroidectomy for PHPT at Severance Hospital between July 2017 and April 2019, 18 were selected and enrolled in the study. The following groups were also excluded from the study: (i) patients who were administered a drug known to affect bone metabolism, such as bisphosphonate, denosumab, calcimimetics, glucocorticoid, thiazolidinedione, thiazide, antidepressant, lithium, aromatase inhibitor, and estrogen at the time of blood collection; (ii) patients who did not undergo dual-energy X-ray absorptiometry (DXA) for BMD measurement 1 year after parathyroidectomy; (iii) patients who had samples with hemolysis ratio higher than 7 or visible hemolysis; (iv) patients who had a history of liver failure, chronic kidney disease stage 4 and above, metabolic bone disease, and other uncontrolled endocrine diseases affecting bone metabolism; (v) patients who had a family history of PHPT or multiple endocrine neoplasia 1; (vi) patients who had experienced any fractures in the last 6 months; and (vii) patients with atypical parathyroid adenoma or parathyroid carcinoma confirmed in the histopathologic report.

BMD was measured using (DXA) (Discovery W, Hologic Inc., Bedford, MA, USA) at the lumbar spine (LS), femoral neck (FN), and total hip (TH) at baseline and post 1 year after parathyroidectomy. The coefficients of variation (CVs) in our institution for areal BMD at LS, FN, and TH were 1.2%, 2.1%, and 1.7%, respectively. The primary outcome of the study was the association of each osteoporotic fracture-related miRNA with the percentage change in TH BMD (%) at 1 year after surgery for the following reasons: (i) there is a greater reduction of cortical bone than cancellous bone in patients with PHPT [26,27,28,29]; (ii) TH BMD is one of the regions that improves 1 year after parathyroidectomy [30]; and (iii) CVs for TH BMD are lower than that for FN in our hospital, which is consistent with the reported literature (CV for TH BMD, 0.8–1.7%; CV for FN BMD, 1.1–2.2%) [31]. The association between each miRNA and percentage changes in LS BMD and FN BMD from baseline, 1 year after surgery, were analyzed as secondary outcomes. The serum CTx level was measured using an electrochemiluminescence immunoassay (Elecsys β-CrossLaps; Roche Diagnostics, Rotkreuz, Switzerland; intra-assay CV < 3.5%, inter-assay CV < 8.4%). PTH and P1NP levels were also measured using an electrochemiluminescence immunoassay (ECLIA; Roche Diagnostics, Rotkreuz, Switzerland; PTH intraassay CV < 2.5%, PTH inter-assay CV < 5.7%; P1NP intra-assay CV < 3.6%, inter-assay CV < 3.9%).

Osteoporosis, osteopenia, and normal BMD were classified according to the International Society of Clinical Oncology guidelines [32]. The T-score was applied for postmenopausal women and men aged 50 years and older, and Z-score was applied for premenopausal women and men aged < 50 years.

### 2.2. Circulating miRNA Analysis

Overnight fasting blood samples were collected 0–2 days before surgery and stored at −80 °C in the Yonsei University College of Medicine. Selected osteoporotic fracture-related miRNAs were extracted and amplified from plasma according to the manufacturer’s instructions and methods, as described in the literature [33,34].

The miRNeasy Serum/Plasma Kit (Qiagen) was used for purification of cell-free total RNA. Briefly, plasma samples were thawed on ice and 1 mL lysis reagent was added to 200 µL sample. Isolated RNA was used for reverse transcriptase polymerase chain reaction (RT-PCR) using the miRCURY LNA RT Kit (Qiagen). The isolated RNA was thawed and mixed with 5× miRCURY RT reaction buffer followed by 10× miRCURY RT Enzyme. To adjust each template RNA to 5 ng/μL, we used 2 μL of the eluate for 200 μL of the plasma samples. After mixing the reverse transcription master mix on ice, the samples were incubated for 60 min at 42 °C and then incubated for 5 min at 95 °C to inactivate the reverse transcriptase enzyme. After cooling to 4 °C, RT-PCR was performed. PCR conditions were: 95 °C for 2 min, 40 cycles of 95 °C for 10 s, and 56 °C for 1 min.

Sixteen circulating miRNAs (hsa-miR-19b-3p, hsa-miR-21-5p, hsa-miR-23a-3p, hsa-miR-23a-5p, hsa-miR-24-2-5p, hsa-miR-24-3p, hsa-miR-93-5p, hsa-miR-100-5p, hsa-miR-122-5p, hsa-miR-124-3p, hsa-miR-125b-5p, hsa-miR-148a-3p, hsa-miR-152-3p, hsa-miR-335-5p, hsa-miR-375, and hsa-miR-532-3p), which were reported to be upregulated in the serum of patients with osteoporotic fracture, were selected [24,33].

The 5s ribosomal RNA (5s rRNA) and miR-16 were used as housekeeping genes, and the stability of the reference gene transcripts was calculated using Delta Ct, BestKeeper, Normfinder, and Genorm algorithms [35,36,37,38]. The 5s rRNA gene was identified as a stable housekeeping gene. All analyses were based on the baseline value (2^−^^∆Ct^). Ct values were normalized to the internal control, and miRNA expression was presented as ∆Ct = (Ct gene of interest—Ct internal control). If the miRNA Ct was ≥40, the value was considered undetermined and was set to ∆Ct = 40. Samples with a hemolysis ratio (calculated by Ct_hsa-miR-23a-3p_ minus Ct_hsa-miR-451a_) of higher than 7 were excluded from the final analysis [33].

### 2.3. Statistical Analysis

Logarithmic transformation (log 2) was used for miRNA analysis. The paired-sample *t*-test and Wilcoxon signed-rank test were used to compare continuous variables, as appropriate. Univariate and multivariate regression analyses were performed to evaluate the association of each miRNA with BMD gain at 1 year after adjustment for age, sex, and baseline PTH level. All statistical analyses were performed using STATA 16.0 (Stata Corp LP., College Station, TX, USA). A two-sided *p*-value of < 0.05 was considered statistically significant.

### 2.4. Target Gene Predictions

Potential target mRNAs for the selected miRNAs were identified via bioinformatics using four different algorithms within the miRWalk3.0 database, namely miRWalk, TargetScan, miRDB, and miRTarBase [39].

## 3. Results

### 3.1. Clinical Characteristics of Study Subjects

A total of 12 patients with PHPT (10 women, mean age 55.2 years) were included in the final analysis (Figure 1). Of the 10 women, 7 were postmenopausal and 3 pre-menopausal. Among the two men, one was above 50 years of age and the other below 50 years of age. According to baseline BMD, three patients had osteoporosis, four patients had osteopenia, and five patients had normal BMD (Table 1). Each patient underwent surgical treatment for the following reasons. Four patients were <50 years old, seven patients had serum calcium levels > 1 mg/dL compared to the normal range, two patients had estimated glomerular filtration rate (eGFR) < 60, one patient had nephrolithiasis, and five patients had a T-score of −2.5 or less. Among the five patients with a T-score of −2.5 or less, two had a forearm T-score of −2.5 or less. One patient was provided surgical treatment because she wanted surgical treatment, and bone loss was progressed. All patients were successfully cured of PHPT as PTH and corrected Ca levels improved to normal ranges within 1–3 days after surgery (median PTH, 109.7 to 40.6 pg/mL, *p* < 0.001; median corrected calcium, 11.0 to 9.1 mg/dL, *p* = 0.001). P1NP, CTx, and 25-OH-vitamin D were also significantly recovered 2 weeks after surgery (median P1NP, 72.5 to 91.8 ng/mL, *p* = 0.003; median CTx, 0.665 to 0.390 ng/mL, *p* = 0.002; mean 25-OH-vitamin D, 19 to 27 ng/mL, *p* = 0.001). Serum calcium levels were maintained in the normal range for more than 6 months after surgery, and we decided that PHPT patients were cured.

### 3.2. Correlation between miRNAs and Study Covariates

The relationships between age, PTH, CTx, P1NP, and each miRNA are shown as a heat map (Appendix A). Although none of the miRNAs were significantly associated with age, baseline PTH, CTx, and P1NP (*p* > 0.05), there was a high correlation between most of the miRNAs, indicating that the selected miRNAs in this study were indeed involved in bone metabolism. All miRNAs, except miR-122-5p and miR-124-3p, showed statistically significant associations with each other (*p* < 0.05).

### 3.3. Higher Preoperative Circulating miR-122-5p and miR-375 Expression Levels Were Associated with Slow Recovery of TH BMD in Patients with PHPT

The association between each miRNA and postoperative TH BMD percent change from baseline, 1 year after parathyroidectomy, is described in Table 2. In univariable regression, miR-19b-3p (β coefficient = −1.55, *p* = 0.038), miR-122-5p (β coefficient = −1.61, *p* = 0.034), and miR-375 (β coefficient = −2.12, *p* = 0.002) were negatively associated with change in TH BMD. After adjustment for age, sex, PTH, and P1NP (Table 3, model III), the association among miR-122-5p, miR-375, and TH BMD changes was significant, whereas no association was observed between miR-19b-3p and TH BMD changes. According to the pre-specified panel, miR-122-5p and miR-375 commonly targeted runt-related transcription factor 2 (RUNX2) in the predicted target model (Appendix A). There was no association between each miRNA and LS BMD or FN BMD changes after surgery in patients with PHPT (Appendix A).

### 3.4. Patients with High miR-122-5p Levels Had Less CTx Change after Parathyroidectomy

In patients with higher preoperative miR-122-5p levels, the postoperative CTx change (%) from baseline, 2 weeks after surgery, was smaller (*p* = 0.009). There was no difference in the postoperative P1NP change (%) from baseline with respect to miR-122-5p levels (*p* = 0.310). There was no difference in CTx and P1NP changes after surgery with respect to miR-375 levels (*p* > 0.05) (Figure 2).

### 3.5. Association between Known Predictors of BMD Response and Postoperative BMD Changes in Patients with PHPT

Predictive markers related to BMD change after parathyroidectomy in patients with PHPT, such as corrected calcium, bone turnover markers (CTx, P1NP), PTH, and age were used in the analysis. Upon conducting the Spearman correlation analysis, corrected calcium was associated with postoperative TH BMD changes (ρ = 0.608, *p* = 0.036). CTx and P1NP were associated with postoperative FN BMD changes (CTx, ρ = 0.587, *p* = 0.045; P1NP, ρ = 0.587, *p* = 0.045). PTH, age, and sex were not significantly correlated with postoperative BMD changes (*p* > 0.05).

## 4. Discussion

In this proof-of-concept study, we investigated whether osteoporotic fracture-related miRNAs were associated with the degree of postoperative BMD changes in patients with PHPT. Previous studies reported potential markers associated with BMD changes after parathyroidectomy in PHPT patients, but the results are inconsistent among studies. Lumachi et al. suggested that premenopausal women had a greater LS BMD (L2-4) gain than postmenopausal women [40]. Koumakis et al. reported a correlation between ALP and BMD improvement [8]. Alonso et al. reported that PTH, CTx, and P1NP levels were associated with changes in LS BMD in simple regression analysis in 53 PHPT patients [9]. By contrast, Steinl et al. suggested that preoperative serum calcium levels, total parathyroid weight on computed tomography, sestamibi avidity, and time from pre- to post-operative DXA are associated with LS BMD improvement after surgery, but not PTH levels [41]. Owing to these controversies, we evaluated whether miRNAs can be used as new markers for postoperative BMD recovery in PHPT patients.

The selected osteoporotic fracture-related miRNAs in our study are known to be involved in bone metabolism; miR-19b-3p and miR-335-5p stimulate osteogenic differentiation [42,43], whereas miR-23a, miR-24-3p, miR-100-5p, miR-125b-5p, and miR-532-3p inhibit osteogenic differentiation [44,45,46,47,48]. Mir-124-3p inhibits osteoclastogenesis by suppressing the nuclear factor of activated T cells 1 (NFATc1) and the receptor activator of nuclear factor k-B ligand (RANKL)-mediated osteoclast differentiation and osteoblastogenesis, by suppressing distal-less (Dlx) [49,50]. MiR-148-3p promotes both osteoclast differentiation via V-maf musculoaponeurotic fibrosarcoma oncogene homolog B (MAFB) and osteogenic differentiation via lysine-specific demethylase 6b (Kdm6b) [51,52]. Inhibition of miR-152 promotes osteoblast differentiation [53]. In the present study, most of the selected miRNAs showed a strong correlation with each other, suggesting that they were involved in similar processes of bone metabolism.

Among the selected osteoporotic fracture-related miRNAs, we found that high miR-122-5p and miR-375 levels were associated with lower postoperative TH BMD changes independent of age, sex, PTH, and P1NP. This is consistent with recent clinical studies that have revealed that miR-122-5p and miR-375 are associated with bone deterioration. miR-122-5p was upregulated in 15 postmenopausal women with hip fracture compared to 12 postmenopausal women with osteoarthritis [54], and was also upregulated in 45 osteoporotic patients with fracture compared to 15 non-osteoporotic patients with fracture [55]. In a study of 30 osteoporosis and 30 non-osteoporosis patients, miR-122a had the highest risk of osteoporosis, with the area under the curve of 0.77 among miR-21, miR-23a, miR-24, miR-93, miR-100, miR-122a, miR-124a, miR-125b, and miR-148a [24]. Meanwhile, Mandourah et al. reported that miR-122-5p is downregulated in osteoporosis (with or without fracture) in a study of 139 patients older than 40 years, which is inconsistent with our study [56]. This difference may result from the inclusion of a small number of patients with fracture or steroid exposure 2 years to 1 month prior to the time of study registration in the study of Mandourah et al. [56]. miR-375 was upregulated in 26 patients with low BMD and vertebral fractures compared to 42 healthy postmenopausal women [33]. In a study of 50 osteoporotic and 49 non-osteoporotic women, high levels of miR-375 were associated with a high risk of osteoporosis [57].

According to target gene predictions (Appendix A), miR-122-5p and miR-375 commonly target RUNX2, which is necessary for osteoblast differentiation [39,58]. Patients with relatively high levels of miR-122-5p and miR-375 might have decreased RUNX2 levels, which might adversely affect osteoblast differentiation after parathyroidectomy in patients with PHPT. Several studies provide evidence that miR-122-5p and miR-375 are associated with RUNX2 and osteoblast differentiation. In an in vitro study, miR-375 inhibited osteogenic differentiation by targeting RUNX2, and this result is consistent with that of our study [59]. Sun et al. demonstrated that miR-375 suppresses osteogenesis by targeting low-density lipoprotein receptor-related protein 5 (LRP5) and β-catenin [60]. However, the effect of miR-122-5p on osteoblast differentiation remains controversial. Meng et al. demonstrated that miR-122 inhibits osteoblast proliferation by activating the Purkinje cell protein 4 (PCP4)-mediated c-Jun NH2-terminal kinase (JNK) pathway [61]. By contrast, bone marrow mesenchymal stem cell-derived exosomes carrying miR-122-5p promote osteoblast proliferation in osteonecrosis of the femoral head, regulating Sprouty RTK Signaling Antagonist 2 (SPRY2) via the RTK/Ras/mitogen-activated protein kinase (MAPK) signaling pathway [62]. The relationship between miR-122-5p and RUNX2 has not yet been investigated.

Interestingly, in this study, PHPT patients with high miR-122-5p levels had significantly less change in CTx 2 weeks after surgery. Patients with high miR-122-5p levels had relatively high CTx levels within two weeks of surgery, adversely affecting BMD recovery. CTx is cleaved by osteoclasts during bone resorption, which suggests a link between miR-122-5p and osteoclasts [63]. Kelch et al. reported that miR-122-5p is upregulated in osteoporotic osteoclasts during osteoclast differentiation compared to non-osteoporotic cells [64], which is in close agreement with our findings. However, the direct effect of miR-122-5p on osteoclastogenesis has not yet been documented. In addition, it may be an effect of RUNX2 degradation. RUNX2 has been suggested to be a major modulator of osteoblast differentiation, and, recently, Xin et al. reported that this modulation occurs via the AKT/NFATc1/CTSK axis [65]. MiR-122-5p may reduce both osteoblast and osteoclast differentiation through RUNX2 degradation, although there was no difference in P1NP change with respect to the miR-122-5p levels in the present study. In fact, P1NP change 2 weeks after surgery may reflect the increased procollagen type I synthesis as the PTH decreases. In PHPT patients, continuous PTH elevation increases the osteoblastic population and osteoblasts synthesize procollagen type I, while PTH plays a role in preventing the synthesis of procollagen type 1 in osteoblast itself [66,67]. Therefore, immediately after surgical treatment in PHPT patients, the synthesis of procollagen type 1 is temporarily increased with increased osteoblastic population and decreased action of PTH [66,67]. Consistent with this, in our study, P1NP was increased 2 weeks after surgery.

There are several limitations to this study. The miRNAs examined here were related to TH BMD changes but not to LS BMD or FN BMD changes. The number of patients in this study was small owing to the rarity of PHPT. Although PHPT is one of the main causes of secondary osteoporosis, it has an incidence of approximately 80 out of 100,000 (0.08%) in Asians [68]. Despite the small study population, however, the characteristics of the study population were similar to the general characteristics of PHPT, with a ratio of female-to-male of 5:1, and the mean age of 55 years [69]. In addition, the existing markers associated with BMD changes after surgery, such as calcium, CTx, and P1NP [9,41], were also associated with BMD changes in the present study. There was no control group, and miRNAs investigated in the present study were selected by literature review, increasing the chances of false-positive target identification. The mechanisms by which miR-122-5p and miR-375 adversely affect postoperative BMD recovery in PHPT patients are yet to be confirmed by a functional study.

In conclusion, preoperative circulating miR-122-5p and miR-375 levels were negatively associated with TH BMD change after parathyroidectomy in patients with primary hyperparathyroidism. Our findings suggest that miRNAs have a potential to facilitate individualized treatment of PHPT. This study was a preliminary proof-of-concept study that requires additional validation.

## Figures and Tables

**Figure 1 diagnostics-11-01704-f001:**
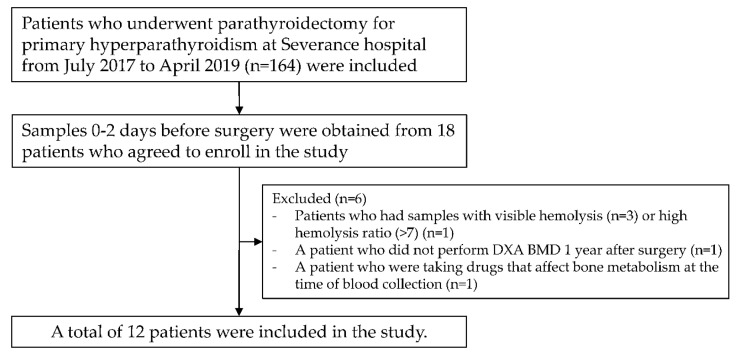
Outline of the study design.

**Figure 2 diagnostics-11-01704-f002:**
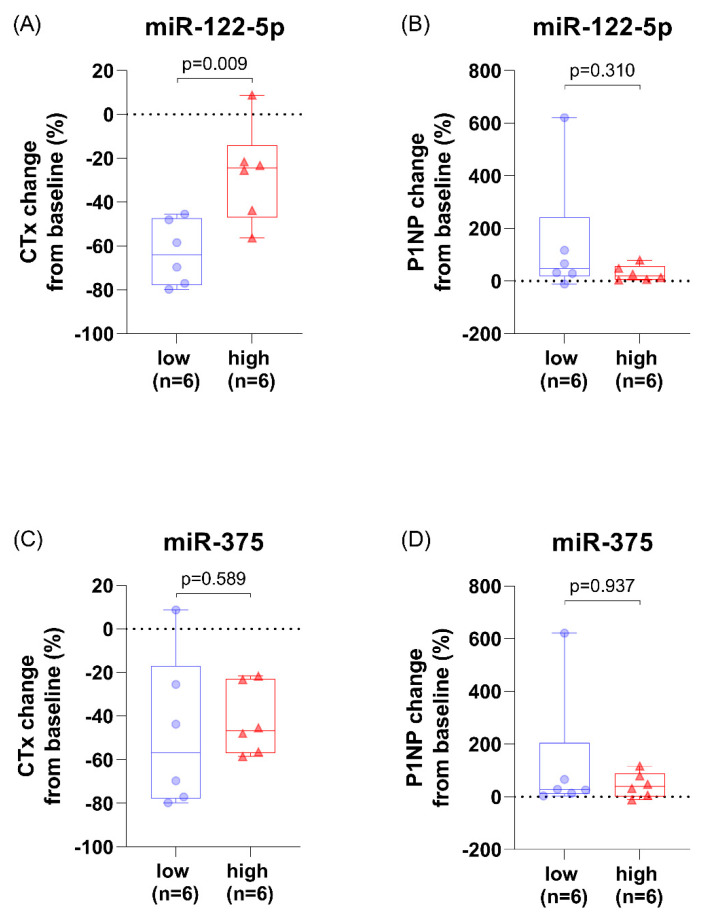
Change in bone turnover markers from baseline (%), 2 weeks after parathyroidectomy, with respect to miR-122-5p and miR-375 levels in patients with PHPT. (**A**) The postoperative CTx change (%) with respect to miR-122-5p levels. (**B**) The postoperative P1NP change (%) with respect to miR-122-5p levels. (**C**) The postoperative CTx change (%) with respect to miR-375 levels. (**D**) The postoperative P1NP change (%) with respect to miR-375 levels. Abbreviations: C-terminal telopeptide of type 1 collagen (CTX) and procollagen type 1 N propeptide (P1NP).

**Table 1 diagnostics-11-01704-t001:** Baseline characteristics.

Variables	Total (n = 12)
% Female (n)	83.3% (10)
Age (years)	55.2 ± 14.9
Body mass index, kg/m^2^	24.3 ± 3.5
Osteoporosis/Osteopenia/Normal, n (%)	3 (25%)/4 (33%)/5 (42%)
Laboratory tests (reference value)	
P1NP, ng/mL (23–83)	72.5 (44.6–100.6)
CTX, ng/mL (<0.573)	0.665 (0.406–1.125)
PTH, pg/mL (15–65)	109.7 (96.9–157.8)
25(OH)D, ng/mL (>20)	18.9 ± 5.5
Corrected Ca, mg/dL (8.5–10.5)	11.6 (9.8–12.4)
Phosphate, mg/dL (2.8–4.5)	2.9 (2.6–3.1)
ALP, IU/L (52–133)	89.8 ± 23.0
eGFR, mL/min/1.73 m^2^ (≥60)	104 (76–110)
Urinary Ca, mg/24 h (70–180)	227 ± 105

Data are presented as frequency (%), mean ± standard deviation, and median (interquartile ranges). Abbreviations: P1NP, procollagen type 1 N-terminal propeptide; CTX, C-terminal telopeptide; PTH, parathyroid hormone; 25(OH)D, 25-hydroxyvitamin D; Ca, calcium; ALP, alkaline phosphatase; eGFR, estimated glomerular filtration rate calculated by EPI.

**Table 2 diagnostics-11-01704-t002:** Univariable regression (β coefficient) for the association of log-transformed preoperative miRNAs (−∆Ct) with total hip bone mineral density (TH BMD) percent change from baseline (%), 1 year after parathyroidectomy, in patients with primary hyperparathyroidism (PHPT).

Change in TH BMD from Baseline (%)—1 Year after Surgery	β Coefficient	*p*-Value
hsa-miR-19b-3p	−1.55	0.038 *
hsa-miR-21-5p	−1.11	0.282
hsa-miR-23a-3p	−0.69	0.447
hsa-miR-23a-5p	0.22	0.893
hsa-miR-24-2-5p	−0.58	0.432
hsa-miR-24-3p	−0.85	0.274
hsa-miR-93-5p	−1.33	0.113
hsa-miR-100-5p	−1.02	0.273
hsa-miR-122-5p	−2.12	0.002 *
hsa-miR-124-3p	1.94	0.319
hsa-miR-125b-5p	−1.14	0.395
hsa-miR-148-3p	−1.26	0.18
hsa-miR-152-3p	−1.37	0.227
hsa-miR-335-5p	−0.68	0.561
hsa-miR-375	−1.61	0.034 *
hsa-miR-532-3p	−0.77	0.463

* indicates a significant association between miRNAs and changes in BMD.

**Table 3 diagnostics-11-01704-t003:** Multivariable regression models for TH BMD change from baseline (%), 1 year after parathyroidectomy, in patients with PHPT.

Change in TH BMD from Baseline (%)	Model I		Model II		Model III	
Variables	β coefficient (95% CI)	*p*-value	β coefficient (95% CI)	*p*-value	β coefficient (95% CI)	*p*-value
miR-19b-3p	−1.55 (−2.99 to −0.10)	0.038	−1.31 (−2.71 to 0.08)	0.062	−1.08 (−2.26 to 0.10)	0.067
miR-122-5p	−2.12 (−3.29 to −0.95)	0.002	−3.00 (−4.56 to −1.45)	0.002	−2.60 (−3.88 to −1.32)	0.003
miR-375	−1.61 (−3.08 to −0.15)	0.034	−1.62 (−2.80 to −0.44)	0.013	−1.34 (−2.25 to −0.43)	0.011

Model 1: No adjustment. Model 2: Adjusted for age and sex. Model 3: Adjusted for age, sex, PTH, and P1NP. Abbreviations: 95% CI, 95% confidence interval; PTH, parathyroid hormone transformed by natural logarithm; P1NP, procollagen type 1 N-terminal propeptide.

## Data Availability

The data presented in this study are available on request from the corresponding author.

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
