# Peer review of "Circulating miR-122-5p and miR-375 as Potential Biomarkers for Bone Mass Recovery after Parathyroidectomy in Patients with Primary Hyperparathyroidism: A Proof-of-Concept Study"

_diagnostics, 2021, doi:10.3390/diagnostics11091704_

Round 1

Reviewer 1 Report

I think we as surgeons focus on the parathyroid surgery but not on its long term effects and benefits.  This research reveals a promising methodology to assess the positive skeletal impact of doing parathyroidectomy and may offer objective criteria for deciding on use of bone therapeutic agents post operatively.   

Congratulations on this large amount of data you will contribute to the parathyroid and osteoporosis literature.  

Author Response

Reviewer’s Comments

Reviewer #1 : I think we as surgeons focus on the parathyroid surgery but not on its long term effects and benefits.  This research reveals a promising methodology to assess the positive skeletal impact of doing parathyroidectomy and may offer objective criteria for deciding on use of bone therapeutic agents post operatively.  

Congratulations on this large amount of data you will contribute to the parathyroid and osteoporosis literature.

Response to reviwer 1: Thank you very much. I appreciate the time and effort that you have dedicated to providing your valuable feedback on my manuscript.

Reviewer 2 Report

This is a very novel and interesting study investigating the potential role of miRNAs as biomarkers for predicting BMD recovery after surgical management of primary hyperparathyroidism (PHPT). Although this is just a proof-of-concept study now, the results suggested that osteoporotic fracture-related miRNAs, especially miR-122-5p and miR-375, may be used for predicting postoperative BMD change in PHPT patients after additional validation.

There are some minor points to consider for revision.

1. In the Results, can the authors clarify the reasons for parathyroidectomy in 12 PHPT patients? Information on indications of parathyroidectomy (e.g. age, calcium level, osteoporosis..) might be helpful for understanding the characteristics of this study subjects.

2. In the Results, when did the authors decide that the PHPT patients were cured? Please provide when PTH and calcium levels were measured for deciding cure of PHPT.

3. In the Results, did the authors also measure BMD at forearm area? Since the association of miRNAs levels with BMD were inconsistent between total hip and femoral neck area, the association of miRNAs levels with forearm BMD might be helpful for increasing the reliability of this study.

4. In the line 184, change “Table 2” to “Table 3”.

5. In the Discussion, line 273-289, can the authors briefly explain the mechanism why the median P1NP levels increased after parathyroidectomy in this study subjects, even though miR-122-5p and miR-375 showed no significant correlation with postoperative P1NP change? This additional discussion will be helpful to understand the role of osteoblast action in BMD recovery after parathyroidectomy in PHPT patients.

Author Response

Response to reviwer 2: Thank you for insightful comments which helped us to improve our manuscript. We tried to revise the manuscript according to your comments, as thorough as possible.

Comment 1: In the Results, can the authors clarify the reasons for parathyroidectomy in 12 PHPT patients? Information on indications of parathyroidectomy (e.g. age, calcium level, osteoporosis..) might be helpful for understanding the characteristics of this study subjects.

Response 1: Thank you for this suggestion. Each patient underwent surgical treatment for the reasons presented in the table below. Four patients were < 50 years old, seven patients had serum calcium levels > 1 mg/dL compared to the normal range, two patients had estimated glomerular filtration rate (eGFR) < 60, one patient had nephrolithiasis, and five patients had a T-score of -2.5 or less. Among the five patients with a T-score of -2.5 or less, 2 had a forearm T-score of -2.5 or less. One patient (patient 5) was provided surgical treatment because she wanted surgical treatment, and bone loss was progressed. According to your suggestion, we addressed this issue in the results section (page 4, line 152-157, red-colored)

Table Surgical indications of parathyroidectomy in each patient.

Patient number

Age < 50

Serum calcium >1 mg/dL above normal

eGFR < 60

24 h urine calcium > 400 mg/day

nephrolithiasis

DXA ≤ -2.5

1

+

+

+

2

+

+

3

+

+

4

+

5

6

+
(radius)

7

+

8

+

9

+

+

(radius)

10

+

+

+

11

+

12

+

+

Comment 2: In the Results, when did the authors decide that the PHPT patients were cured? Please provide when PTH and calcium levels were measured for deciding cure of PHPT.

Response 2: Thank you for your comment. PTH and calcium levels of the patients were normalized within 1-3 days, and P1NP, CTx, and 25-OH-vitamin D levels were recovered 2 weeks after surgery. Serum calcium levels were maintained in the normal range for more than 6 months after surgery, and we decided that PHPT patients were cured. According to your comment, we addressed this issue in the results section (page 4, line 157-164, red-colored)

Comment 3: In the Results, did the authors also measure BMD at forearm area? Since the association of miRNAs levels with BMD were inconsistent between total hip and femoral neck area, the association of miRNAs levels with forearm BMD might be helpful for increasing the reliability of this study.

Response 3: Thank you for the suggestion. Forearm BMD was performed in 9 patients both before and after surgery, and unfortunately, there was no correlation between change in forearm BMD from baseline (%) 1 year after surgery and selected osteoporotic fracture-related miRNAs. It is likely due to the small number of samples (Table below). Since forearm BMD was not measured in all study populations and the number was small, the association between forearm BMD and selected miRNAs was not included in the results.

Table Univariable regression (β coefficient) for the association of log-transformed preoperative miRNAs (-â–³Ct) with forearm bone mineral density percent change from baseline (%) 1 year after parathyroidectomy in patients with primary hyperparathyroidism (PHPT).

Change in forearm BMD from baseline (%) - 1 year after surgery

β coefficient

p-value

hsa-miR-19b-3p

-0.491

0.534

hsa-miR-21-5p

0.238

0.830

hsa-miR-23a-3p

0.280

0.777

hsa-miR-23a-5p

-0.118

0.940

hsa-miR-24-2-5p

0.095

0.898

hsa-miR-24-3p

-0.053

0.948

hsa-miR-93-5p

-0.509

0.568

hsa-miR-100-5p

0.546

0.628

hsa-miR-122-5p

-0.024

0.977

hsa-miR-124-3p

1.899

0.326

hsa-miR-125b-5p

1.256

0.373

hsa-miR-148-3p

-0.442

0.656

hsa-miR-152-3p

0.081

0.950

hsa-miR-335-5p

0.547

0.664

hsa-miR-375

0.230

0.785

hsa-miR-532-3p

0.316

0.773

Comment 4: In the line 184, change “Table 2” to “Table 3”.

Response 4: Thank you for the nice reminder. We changed “Table 2” to “Table 3” (page 5, line 190-192, red-colored).

Comment 5: In the Discussion, line 273-289, can the authors briefly explain the mechanism why the median P1NP levels increased after parathyroidectomy in this study subjects, even though miR-122-5p and miR-375 showed no significant correlation with postoperative P1NP change? This additional discussion will be helpful to understand the role of osteoblast action in BMD recovery after parathyroidectomy in PHPT patients.

Response 5: Thank you for the comment. In PHPT patients, procollagen I synthesis is increased due to the increased osteoblastic population caused by continuous PTH stimulation, while PTH plays a role in preventing the synthesis of procollagen type 1 in osteoblast itself [1,2]. Therefore, immediately after surgical treatment of PHPT patients, the synthesis of procollagen type 1 is temporarily increased with increased osteoblast population and decreased action of PTH [1,2]. Consistent with this, in our study, P1NP was also increased 2 weeks after surgery. Meanwhile, CTx is known to decrease immediately after surgery because the new remodeling cycle induced by PTH disappears, compatible with our study [3]. High miR-122-5p and miR-375 levels are likely to be involved in RUNX2 degradation and inhibit osteoblast differentiation so that the osteoblastic population may be relatively small and preosteoblastic proliferation and differentiation may be more reduced after surgery [4-8]. However, since the increase in P1NP immediately after surgery reflects the increased procollagen type I synthesis as the PTH decreases, there was no relationship between P1NP and miR-122-5p or miR-375 [2]. Moreover, it was challenging to show the relationship between P1NP and these miRNAs because the number of study population was small, and the existing bone turnover markers had a substantial variation. According to your comment, we addressed this issue in the discussion section (page 9, line 312-319, red-colored)

References

  1. Coen, G.; Mazzaferro, S.; De Antoni, E.; Chicca, S.; DiSanza, P.; Onorato, L.; Spurio, A.; Sardella, D.; Trombetta, M.; Manni, M. et al. Procollagen type 1 c-terminal extension peptide serum levels following parathyroidectomy in hyperparathyroid patients. Am J Nephrol 1994, 14, 106-112.
  2. Guo, C.Y.; Holland, P.A.; Jackson, B.F.; Hannon, R.A.; Rogers, A.; Harrison, B.J.; Eastell, R. Immediate changes in biochemical markers of bone turnover and circulating interleukin-6 after parathyroidectomy for primary hyperparathyroidism. Eur J Endocrinol 2000, 142, 451-459.
  3. Christiansen, P.; Steiniche, T.; Brixen, K.; Hessov, I.; Melsen, F.; Heickendorff, L.; Mosekilde, L. Primary hyperparathyroidism: Short-term changes in bone remodeling and bone mineral density following parathyroidectomy. Bone 1999, 25, 237-244.
  4. Sticht, C.; De La Torre, C.; Parveen, A.; Gretz, N. Mirwalk: An online resource for prediction of microrna binding sites. PLoS One 2018, 13, e0206239-e0206239.
  5. Komori, T. Regulation of osteoblast differentiation by runx2. In Osteoimmunology, Springer: 2009; pp 43-49.
  6. Sun, T.; Li, C.T.; Xiong, L.; Ning, Z.; Leung, F.; Peng, S.; Lu, W.W. Mir-375-3p negatively regulates osteogenesis by targeting and decreasing the expression levels of lrp5 and β-catenin. PLoS One 2017, 12, e0171281.
  7. Du, F.; Wu, H.; Zhou, Z.; Liu, Y.U. Microrna-375 inhibits osteogenic differentiation by targeting runt-related transcription factor 2. Exp Ther Med 2015, 10, 207-212.
  8. Meng, Y.C.; Lin, T.; Jiang, H.; Zhang, Z.; Shu, L.; Yin, J.; Ma, X.; Wang, C.; Gao, R.; Zhou, X.H. Mir-122 exerts inhibitory effects on osteoblast proliferation/differentiation in osteoporosis by activating the pcp4-mediated jnk pathway. Mol Ther Nucleic Acids 2020, 20, 345-358.